# 'Bringing testing closer to you': barriers and facilitators in implementing HIV self-testing among Filipino men-having-sex-with-men and transgender women in National Capital Region (NCR), Philippines – a qualitative study

Jeanno Lorenz G Dinglasan [1], John Danvic T Rosadiño [1,2]
Ronivin G Pagtakhan,[1] Denis P Cruz [1,2] Matthew T Briñes,[1,3]
Zypher Jude G Regencia [4,5] Emmanuel S Baja [4,5]

For numbered affiliations see end of article.

**Correspondence to**
Dr Emmanuel S Baja;
esbaja@up.edu.ph

## ABSTRACT

**Objectives** Our study identified barriers and facilitators in implementing HIV self-testing (HIVST), including the perceptions of men-having-sex-with-men (MSM) and transgender women (TGW) on HIVST. Furthermore, we explored the current knowledge, practices and potential of HIVST among the MSM and TGW populations.

**Design** Qualitative in-depth key informant interviews were administered using semistructured interviews administered in both English and Filipino. Thematic analysis of the findings was done after transcribing all audio recordings.

**Setting** The study was done in the National Capital Region (NCR), Philippines using online video conferencing platforms due to mobility restrictions and lockdowns caused by the COVID-19 pandemic.

**Participants** All study participants were either MSM or TGW, 18–49 years old and residing/working in NCR. Exclusion criteria include biologically born female and/ or currently on pre-exposure prophylaxis, antiretroviral therapy medications or an HIV-positive diagnosis.

**Results** Twenty informants were interviewed, of which 75% were MSM, and most of them preferred the use of HIVST. Facilitators and barriers to the use of HIVST were grouped into three main themes: Acceptability, distribution and monitoring and tracking. Convenience and confidentiality, overcoming fears and normalisation of HIV testing services (HTS) in the country were the participants' perceived facilitators of HIVST. In contrast, lack of privacy and maintenance of confidentiality during kit delivery were perceived as barriers in HIVST implementation. Moreover, social media was recognised as a powerful tool in promoting HIVST. The use of a welcoming tone and positive language should be taken into consideration due to the prevalent HIV stigma.

**Conclusions** The identified facilitators and barriers from the study may be considered by the Philippine HTS programme implementers. The HIVST strategy may complement the current HTS. It will be very promising to involve the MSM and TGW communities and other key

## Strengths and limitations of this study

► Gathering participants and building trust to give their insights did not pose any difficulty because the community-based organisation is reputable among the key populations (men-having-sex-with-men (MSM) and transgender women (TGW)) and HIV advocacy in the Philippines.

► Our study only involved respondents from the National Capital Region, which may limit its generalisability to other regions of the country where HIV cases are still increasing.

► Key informant interviews were done online due to the COVID-19 pandemic; thus, some reactions to the questions of the participants are not noted due to the unavailability of video and weak internet connectivity.

► Perceived acceptability of HIV self-testing (HIVST) from the informants is suggestive, and many of the participants had not used the HIVST before.

► Our study only involved the MSM and TGW populations and did not include other at-risk and high-risk populations, limiting its applicability to different populations.

populations to know their HIV status by bringing testing closer to them.

## INTRODUCTION

It is estimated that there are 79 million HIV infections, 36 million HIV-related fatalities and 38 million people living with HIV.[1] Following the rapid and continued increase in new diagnoses since 2007, the Philippines has been recognised as having one of the fastest-growing HIV epidemics in the world.[2] Furthermore, the HIV epidemic in the

country has been rapidly changing and expanding in the last 5 years, in which 32 cases a day have been diagnosed.[3] There were 62 029 HIV cases from 1984 to 2018, of which 93% were males and 51% were from the age group of 25–34 years old.[4]

The limited onward transmission of HIV can be achieved through early awareness of HIV testing.[5] It has been demonstrated in a randomised controlled trial wherein a 96% reduction in HIV transmission was associated with an early diagnosis, which is a requirement to initiate treatment.[6] However, it must be emphasised that the reduced onward HIV transmission due to awareness in HIV testing must be coupled with rigorous strategies to get these informed individuals engaged in testing. Furthermore, various strategies may include government activities partnered with community-based organisations, enlisting community leaders for public events and promoting HIV testing in facility-based settings and community-based events.[7] In addition, this early awareness of testing and diagnosis accentuated the public health concern that late diagnosis of HIV is greatly affected by the health-seeking behaviour of many at-risk individuals.[8 9]

There was a significant increase in HIV testing services (HTS) from 2010 to 2014; more than 600 million people received proper HTS in 122 low-income and middle-income countries.[10] The expansion of HTS in these countries may be attributed to the increased introduction of provider-initiated testing and counselling and different community-based methodologies, which are now considered a standard of care.[11] However, despite these innovative approaches, approximately 40% of all HIV infections remained undiagnosed globally.[12] Therefore, several countries had already sought approaches to encourage people to get tested for HIV to achieve the first of the United Nation's 90–90 HIV testing and treatment goals-diagnosis of 90% of all people with HIV.[13]

The fear of discrimination, stigmatisation, and access to testing remained public health concerns in many regions that persistently threaten the acceptance of HTS in the world.[14] The US Centers for Disease Control and Prevention has suggested HIV screening routinely for patients between ages 13 and 64 since 2006; however, the uptake of these HTS has been obstructed by several barriers.[15 16] Therefore, globally acceptable and easily accessible HTS are desirable to expand the coverage of diagnosis, which will eventually allow for early antiretroviral treatment for infected individuals. In addition, it will also limit HIV transmission to people who are at risk.[6]

Moreover, increasing evidence that HIV self-testing (HIVST) may help reach those unwilling to use HTS in designated centres, increasing the number of people diagnosed with HIV.[17] One study showed that 70% of those who had ever had an HIV test also had self-tested, with privacy being the main reason for self-testing.[18] Consequently, a well-designed and implemented HIVST programme could effectively increase uptake of HIV testing with increased privacy and efficiently increase entry into HIV prevention, care and treatment services.

Additionally, HIVST could require fewer human resources and lower costs than other alternative approaches.[19]

In the Philippines, the Department of Health (DOH) does not have any HIVST policies or guidelines to regulate HIVST kits purchased online.[20] Currently, a limited study demonstrated the feasibility and acceptability of HIVST in the Philippines.[20] Our research focused on identifying barriers and facilitators in implementing HIVST, including the perceptions of men-having-sex-with-men (MSM) and transgender women (TGW) HTS users and non-users on HIVST. Furthermore, we also explored the current knowledge, practices and potential of HIVST among the MSM and TGW populations in the National Capital Region (NCR).

## MATERIALS AND METHODS

The study was carried out through a non-government organisation, LoveYourself (LYS), from March 2020 to May 2020 in NCR, Philippines. Qualitative methods were used to ascertain facilitators and barriers in implementing HIVST, explore the gaps for lack of access to HTS, and investigate HIVST uptake and linkage to further diagnosis and care among MSM and TGW.

### Study participants

All study participants were screened using the following inclusion criteria: either MSM or TGW, 18–49 years old, and residing or working in the NCR. In addition, recruited study participants were grouped for key informant interviews (KIIs) into four clusters, including (1) MSM who have been tested/screened for HIV at least once, (2) TGW who have been tested/screened for HIV at least once, (3) MSM who were never tested for HIV and (4) TGW who were never tested for HIV. Participants who are biologically born female and/or currently on pre-exposure prophylaxis (PrEP), antiretroviral therapy (ART) medications or an HIV-positive diagnosed individual were excluded from the study. Moreover, recruitment and referral of the KII participants were done through online platforms, such as social media and referrals from LYS volunteers. In addition, the identified key informants were active and supportive members of the LGBTQIA +community.

### Data collection

Data collection included qualitative in-depth semi-structured interviews with KII participants that were administered in both English and Filipino. However, due to the COVID-19 pandemic, face-to-face conduct of KIIs was prohibited. Therefore, KIIs were conducted online through different video conferencing software, whichever was available to the participants. Prior to the interviews and discussions, participants were asked to sign an online informed consent form.

Topic guide questions focused on three different segments. The first segment was focused on respondents' knowledge, acceptability of using HIVST, referral of

HIVST to other people, and inclusion of other materials or services to make HIVST preferable, especially to the first-time testers. The second segment included suggestions on the marketing and distribution of HIVST kits. Additionally, it also had the client's preference on the mechanism of getting samples (blood or oral fluids), on requesting and obtaining the kits in various platforms, on the desired price, and on how to market HIVST and reach the MSM and TGW users. The last segment focused on monitoring and tracking the results, including post-counselling services, follow-up and referral mechanisms for reactive and non-reactive results. Furthermore, other concerns and integration of mental health services primarily to clients who turn out reactive after using the kit were also incorporated in the last segment.

Gathering and analysing data continued until saturation was reached. It was done to develop a robust, valid, and better in-depth understanding of HIVST from the perceptions of MSM and TGW. Moreover, saturation provided the research team an indication of the validity of the data as well as its quality. All interviews from the participants were audiorecorded and were transcribed afterward. In addition, discussions among the researchers were done post-interview to outline a plan for the data analysis.

### Data analysis

The study used a template analysis framework to code for the qualitative interview data by generating an initial list of themes or deductive codes based on the interview topics and applying these codes to the data. Transcripts of the audio recording were indexed using an analytical framework. Summarised data were inputted within a framework matrix, followed by in-depth analysis and interpretation.

Based on the review of the deductively coded data, inductive codes were used to flag data on themes that emerged within or across the topics. One reviewer primarily did coding following an independent coding exercise, which generated a consensus around a single coding framework. The team used the code reports to discuss the main ideas and explored and charted relationships among the themes. Pseudonyms were used to present quotes from qualitative findings.

### Data protection

The data gathered, including the participant information, were kept confidential and private following the Philippines' Data Privacy Act of 2012. The principal investigator and research team did not disclose the identities of the participants at any time. All study data were recorded, and investigators were responsible for the integrity of the data, such as accuracy, completeness, legibility, originality, timeliness, and consistency.

### Patient and public involvement

Patients or the public were not involved in our research's design, conduct, reporting or dissemination plans.

**Table 1** Sociodemographic characteristics of the key informants (N=20)

| Characteristics | n (%) |
|---|---|
| Gender Identity | |
| MSM | 15 (75) |
| TGW | 5 (25) |
| Age, years | |
| 18–24 | 6 (30) |
| 25–35 | 11 (55) |
| 36–49 | 3 (15) |
| Employment status | |
| Unemployed | 1 (5) |
| Employed | 19 (95) |
| Already tested for HIV (facility/community-based) | 17 (85) |
| Have HIVST experience | 1 (5) |

HIVST, HIV self-testing; MSM, men-having-sex-with-men; TGW, transgender women .

### RESULTS

A total of 20 informants were interviewed, of which 75% were MSM, and the rest were TGW. In terms of age, the age bracket 25–35 years old had the highest proportion (55%). In addition, 1 was unemployed, 1 was a government employee, 16 were working in a private sector (15 full time and 1 part time), and 2 informants were self-employed. Thirteen out of 15 MSMs had tested for HIV, while 4 of the 5 TGWs had already been tested. Moreover, only one informant had ever used a HIVST kit purchased through an online channel (see table 1 for details).

### HIVST themes

Different facilitators and barriers to using HIVST were identified based on the information gathered from the MSM and TGW participants. These included three primary themes: (1) acceptability, (2) distribution and (3) monitoring and tracking (figure 1).

#### Acceptability of HIVST

The knowledge about HIVST, comfort in using the HIVST kit, preference of HIVST or facility-based for HIV screening, precounselling for first-time testers to HIVST service, and the choice between oral fluid-based and blood-based HIVST kits were all documented from the MSM and TGW informants as important factors for the acceptability of HIVST.

##### *Knowledge about HIVST*

Informants learnt about HIVST through HIV/AIDS advocacy volunteers from the testing facilities, online shopping platforms, and social media accounts. In addition, one informant mentioned that this was the first time he had heard about an organisation spearheading this project.

I never heard about this before, but some people I know order HIVST kits online. There are no

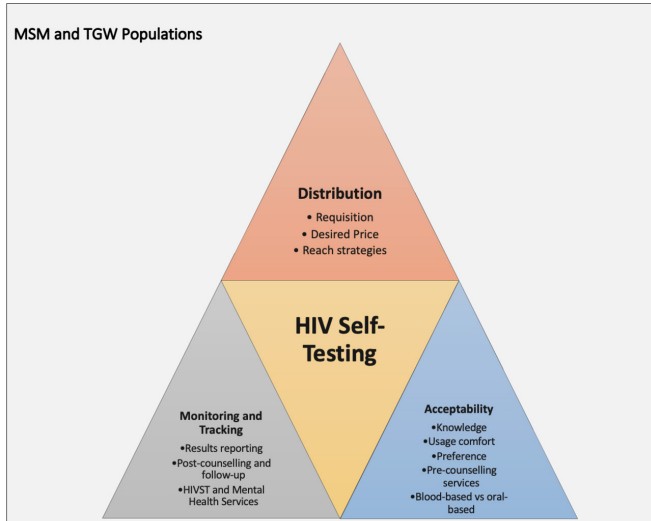

**Figure 1** Identified HIVST themes. HIVST, HIV self-testing; MSM, men-having-sex-with-men; TGW, transgender women.

regulations or monitoring on how to get the kits. They are doing this on their own, and they are the ones self-screening themselves. When it comes to an organisation that will introduce this, this is the first time I heard about this. - [MSM1]

Most of the informants said HIVST is convenient and saves time. In addition, some informants described HIVST as 'discreet' and 'very confidential.'

I've heard about it and find it okay. It will save your time, especially in traveling to visit the center, and good for those who are shy to visit the center. When I had my first test, it was not easy for me to visit the center. The self-testing kit is good. - [MSM2]

In contrast, informants who did not know HIVST said HIVST is convenient but is concerned about the HIVST results and what to do after.

This is helpful because each person has different personalities, and not everyone is brave enough to know their status in a community center… like a person who will use the kit but may not be knowledgeable enough and may not be ready to know their HIV status. - [TGW4]

### Comfort in using the HIVST kits

Most of the informants will be comfortable using the HIVST kit, while others will not. The uncomfortable participants said that readiness to know the result, emotional stability of the person using it, and preference for counselling sessions first before conducting the HIVST influenced their confidence in using the kit.

My concern is about the readiness of the client. Are they strong enough to find out the result? Sometimes, it's still difficult, even if there is a counselor who will guide them. It might lead to something that we do not want to happen. HIVST kit is good, but there

must be a set of qualifications before giving the kit to anyone interested - [MSM2]

### Preference of HIVST or facility-based for HIV screening

Some informants preferred HIVST over facility-based HIV screening for the following reasons: proximity to testing facilities, decongestion of testing centres, and privacy.

I'm looking into the concern of those people who live far away from any testing centers. HIVST will help us reach those people. HIVST can also be offered not only to the LGBTQIA community but also to cisgender people. - [TGW2]

Conversely, informants, who preferred facility-based screening, have the following reasons: the availability of psychosocial help and proper knowledge and assistance provided to the clients, affordability, the accuracy of the result and convenience.

I prefer testing in the facility because there are counselors that will assist me just in case it turns out reactive. The people in the facility will make you feel that you are worthy even if it turns out reactive. - [TGW4]

In addition, HIVST must be offered alongside other screening methods to every client. One informant articulated that he would use HIVST for the sake of his 'sanity,' whereas he will use different methods of screening for validation purposes.

### Precounselling service for first-time HIVST testers

Most participants agreed that pre-counselling should be provided as part of HIVST service for first-timers, while some said it should be optional. In addition, they emphasised that pre-screening counselling is necessary, and a hotline or a website must be provided in case the client wants to seek help on how to use the kit.

We have to take into consideration the emotional and mental health of a person. Know first if they are capable enough to handle the situation and the result. Pre-counseling may be given if they did not meet the requirements needed. However, some people may feel uncomfortable in counseling. It must be optional, and it depends on the preference of the person. - [MSM 14]

At least give a hotline to report the result and educate them on what to do next. It must be included in the instructional manual. Pre-screening counseling may be done; there must be a conversation first. - [TGW1]

### Oral fluid-based versus blood-based HIVST kits

Most participants preferred blood-based HIVST kits than opting for oral fluid-based kits. Accuracy and reliability were the reasons for those participants who chose the blood-based kits. Alternatively, informants who chose oral fluid-based kits were afraid of needles but claimed that the kits were easier to administer.

I got used to blood-based tests because this is what the facilities are using. I have a feeling that it is more accurate than oral fluid-based. - [TGW5]

I would prefer oral fluid-based. Will you just swab it right? I am afraid of pricking myself. Swabbing is easier. I don't see anything preferable when it comes to blood-based. - [MSM6]

### Distribution of HIVST kits

The most efficient and appropriate ways to distribute HIVST kits to respondents in NCR, Philippines, were identified.

#### Requisition of HIVST kit

Most informants said that clients could request an HIVST kit through a website and a mobile application. In addition, other means of ordering an HIVST kit may include the use of social media, community centres with riders, pharmacies and call and/or text-based services.

I'm thinking of the best way for the masses to order the kit. Messaging apps such as Viber and Facebook Messenger are good, but I am thinking of those who do not have access to the internet. Texting is the best way and most convenient for all. - [MSM6]

#### Obtaining HIVST kit and desired price

Most respondents wished to obtain their kits through an online courier service. However, other distribution channels like pharmacies, health facilities, vending machines, partnerships with organisations, and pick-up from an individual staff were also identified.

Courier delivery is the best for me because what I am looking at are accessibility and anonymity… It is better if the kit is delivered to the comfort of your home. Delivery via staff will be good as well. Partnerships with online shopping companies will also be a good option. - [TGW2]

Packaging and delivery of the kits is another concern: how to order the kits, discreet packaging, readability, and friendly tone of the inserts. Furthermore, for the desired price of the HIVST kit, between PHP50 and PHP300 (US$1.00–US$6.00) was the most preferred price range.

The desired price must be at least the price of a pregnancy test kit or at least the price of 3 boxes of condoms. It must be around Php. 150.00. It must be affordable enough to be reached by everyone. For the working class, it must be not more than Php. 1000.00 or 500.00 plus. - [MSM14]

It would be better if it is for free if it is with pay; It must be affordable. - [TGW4]

#### Strategies to reach HIVST targeted population

Most of the informants agreed that social media platforms should be utilised to reach the HIVST targeted population. In contrast, others expressed that help from HIV ambassadors or advocates, schools, private companies,

local government units, mainstream media and online dating sites were also necessary. Furthermore, strategies should focus on maintaining convenience and confidentiality with the HIVST kit users.

…through Social Media… specifically via Twitter or any platform because everyone uses social media. They don't need to visit a center because some people still have the stigma. It lessens the fear of being judged by people. Emphasize the idea of confidentiality and convenience. - [MSM1]

…utilize influencers, I'm pretty sure that everyone will imitate and use it. Urge influencers to use their page and demonstrate how to use the kit. - [MSM3]

Other strategies considered include the maturity, socioeconomic and academic status of the clients, data on the current number of HIV cases in the country, affordability and availability, usage of positive language, the benefits of using HIVST kit, and the treatment process after being diagnosed with HIV.

We may market the kits using the strengths like you can do it at the comfort of your own home and the like. Introduce data on the severity or death tolls of HIV/AIDs. This is not to frighten them, but to create a realization on their part that they must not be part of the statistics. - [MSM6]

### Monitoring and tracking of HIVST usage

Information regarding monitoring and tracking of HIVST service were ascertained from the participants, including (1) reporting of client's result, (2) postcounselling resources for non-reactive and reactive clients, (3) a timeline for visiting the facility for testing and linkage-to-care for reactive clients, (4) follow-up services for those reactive clients to report their result and (5) incorporation of mental health services to HIVST service.

#### Reporting of results

The kits must include maps of facilities and hotlines for the clients to contact if the result turns out to be reactive.

… contact numbers must be provided, and a call center within the organization must be established to address the concerns of the clients. - [TGW2]

An incentive system must be given and regular counseling. There must be a mechanism on how to follow-up the clients. - [TGW1]

Assurance of assistance and postcounselling services were the most appropriate reporting strategies. Highlighting the power of choice, confidentiality and using platforms and mobile applications were some of the other responses given by the informants.

In the website, there must be a statement stating the beauty of reporting the results. They are considered clients the moment they avail themselves of the kit. Inform them that the best way to get complete

healthcare is to report their HIV status. If the client doesn't want to, then we have to respect their choices. One of the reasons why they are afraid of having themselves tested is confidentiality. If this will be ensured, then they will be confident to report their results. - [MSM3]

### Postcounselling services and follow-ups

A postcounselling service for both the non-reactive and reactive clients must be necessary. In addition, follow-ups for the clients who turned out reactive after using HIVST kits must also be administered through call or text messages.

Regardless of the status, we must send a text message first before post-counseling. This is to ensure the client is ready to accept the result for both reactive and non-reactive results. Post-counseling must be done, but it must be upon the approval of the client. - [MSM15]

The counselor must talk to the client beforehand and must give options for the follow-up so that the client will not feel like someone is invading their privacy. This must be introduced during the pre-counseling. - [MSM8]

### Mental health and HIVST services

Support groups must be established by the organisation to address, assist, and improve the mental health of the clients whose tests will turn out reactive. Moreover, the inclusion of contact details of mental health professionals, linkages and continuous counselling to address the client's mental health were all identified.

Psychological support groups are good but not for me (personally); reaching out to professional mental health practitioners may be a better option. Contact details and centers for mental health must also be provided in the kits. - [MSM4]

### DISCUSSION

Currently, there are four programmes against the HIV epidemic: (1) precaution, (2) screening or diagnosing for HIV, (3) antiretroviral treatment (ART) and (4) PrEP.[21] The increase in the incidence of specific diseases is attributed to the population's active participation in disease screening. Private and government hospitals, non-government organisations, and social hygiene clinics (SHCs) offer HIV testing and counselling (HTC) services in the Philippines. However, in a recent report of the Philippine Department of Health, the MSM population still has a record of 78% not being tested for HIV. About 30% of 22% who got tested for HIV reported being comfortable going SHC to avail HTC services.[22] Primary care physicians described the social stigma associated with HIV, the conservative nature of some communities, and the lack of privacy available as barriers to HIV

testing.[23] The need to expand the HTC service options in the Philippines for MSM and TGW is crucial, especially in highly stigmatised environments. HIVST provides a choice to testing aside from the facility-based screening. This complementary approach can increase HTC services uptake and frequency, thereby increasing early HIV detection and prevention of new infections and early linkage to care.[24–27]

Our qualitative study confirmed the acceptability of HIVST among the target populations, specifically those at risk of contracting HIV, the MSMs and the TGWs. Acceptability of HIVST kits among the informants revolved around various strata of concerns, including current knowledge about HIVST, comfort in using kits, preference of HIVST over facility-based screening, and oral-based vs blood-based kits. Informants welcomed establishing an HIVST system and recognised that the system would expand the reach for HIV screening. In most countries, the acceptability of HIVST is high. In Malawi, the acceptability of HIVST offered at home with minimal supervision was noted to be high.[28] Furthermore, at least 67% of the high-risk MSM population in the United States reported the utilisation of HIVST.[29] In Mpumalanga, South Africa, researchers documented high uptake and acceptability of HIVST following the distribution of the kits to MSM populations.[30] In addition, a recent study in Australia demonstrated that populations at-risk for HIV infection tested twice using HIVST compared with the at-risk populations offered with clinic-based screening.[31] The results shown in other countries agreed with our findings on the acceptability of HIVST.

Moreover, the knowledge on HIVST of the participants came from the SHCs, social media and online shopping. Studies in China have demonstrated how online sites provide information on HIVST, including informational videos and basic counselling. The information these sites are gearing was towards the promotion of HIVST.[32–34] Online shopping is also one of the biggest platforms to display information on HIVST in China. However, the purchase of HIVST through online shopping entails a lack of quality assurance and monitoring. The HIVST kit sold online also posits questionable integrity and accuracy.[27] The framework on the checking of quality assurance of HIVST kits being sold online should be developed following the existing rules and regulations of the Philippine Food and Drug Administration to ensure accurate reporting of HIV cases in the Philippines. This approach should be thoroughly considered in the future.

Our study also presented constraints and opportunities in terms of the distribution of the HIVST kit. Social media is the best way to acquire these kits and can be delivered through established courier companies but with the utmost confidentiality. The use of social media has already increased the reach for HIVST, which provided new avenues to target testing among the key populations.[35] Different platforms on smartphones such as Grindr, Facebook, and Twitter have been determined to effectively reach the MSM and TGW populations tested

using HIVST.[36–39] Using social media to promote HIVST facilitated overcoming constraints when being tested in a facility such as privacy or possible leak of data on sexual orientation.[40] In addition, other countries use different online platforms to promote and distribute HIVST kits among MSMs and TGWs, such as WhatsApp in Nigeria, which presented data that there was an increase in the uptake of testing among MSMs when HIVST kits were promoted using WhatsApp.[41] In China, WeChat has been steadily used to deploy HIVST projects; the MSM population in China trusts this messaging platform.[42–44] The possible use of different social media platforms to target MSMs and TGWs in the Philippines to increase uptake and distribution of HIVST kits should be explored immediately.

It is also notable that the distribution of HIVST kits is highly dependent on the possible price of the HIVST kits in the future once it is available. Our study observed a suggested price range for HIVSTK of PHP 50–PHP300 (US$1.00–US$6.00) as preferred by the majority of the respondents. Additionally, the mechanisms of the government to increase the uptake of testing using HIVST can be achieved if a reduced price for the kits is realised. This strategy can be accomplished if manufacturers and sponsors will partner with key health stakeholders in the country. Low costs of HIVST kits will develop confidence about the availability of affordable kits in the future.[45] Past experiences suggested that the HIVST kit costing about US$3 could save US$53 million over 20 years using a cost-effective analysis.[46] Furthermore, an affordable HIVST kit may potentially increase testing among key populations tied up with low-cost interventions among HIV-positive individuals.[47] The wealth of evidence and data presents many factors to be considered in developing algorithms and models to fully implement HIVST in the Philippines; further explorations of these factors are needed to strengthen the future HIVST scale-up implementation.

The gap between the result of HIVST and the services after using the HIVST kit needs to be addressed. The third theme presented in our qualitative study suggested that monitoring and uptake of HTC services after using the kits are essential parameters to be considered. Reporting results should be standardised, and post-counselling services should be made available for reactive and non-reactive clients, highlighting mental health services as the main HIV/AIDS awareness programme. One research described their HIVST reporting algorithm wherein kits were ordered online to be delivered through mail or self-pickup at a chosen pharmacy with a random personal identification number (PIN) to track future distribution and requests. The assigned PIN also anonymised clients to report their results online and monitor subsequent linkage to further HTC services.[48] The need for confirmatory testing, postcounselling services, and immediate linkage to care are important after receiving a reactive HIVST result. Other studies also raised concerns on HIVST, particularly whether post-counselling after testing should be done face-to-face to

not exacerbate negative behaviours and unfavourable consequences.[49 50] Reservations on HIVST have also been expressed, including emotions after a reactive result and misuse of the kits. The lack of supervision and postcounselling is built-in to HIVST, and distress on mental health is possible.[17 51] Therefore, the need for a robust mental health programme complementing HIVST implementation is vital, as verified by the information gathered from the respondents of our study.

The failed attempt of the government to address the constant increase of HIV incidence in the Philippines prompted lawmakers to enhance its current HIV/AIDS law to provide a better response to this epidemic.[52] As the epidemic of HIV persistently grows, the introduction of HIVST to complement facility-based screening might offer a new advantage in increasing the number of people getting tested and being aware of their HIV status. A recent study concluded that strategic cooperation between the Philippine stakeholders to deliver a patient-centred HIVST programme to escalate testing coverage.[20] Our study, which offered substantial data regarding facilitators and barriers to HIVST implementation, may be adopted by policy-makers and HIV programme implementors to introduce a standardised way to cascade HIVST from the national to the local levels. There is no known single protocol to give an accurate and precise way to link HIVST usage and HTC services due to several limitations. However, information regarding identified factors in reducing possible limitations of HIVST implementation can increase the confidence and minimise the possibility of implementation failure, as evident from the data our study presented.

The results of our research stemmed out from the use of a local network of the identified community organisation. This community organisation is popular among the key populations (MSM and TGW) and in HIV advocacy in the country; thus, trust between the participants and the researchers to give insights regarding HIVST has already been built.

## Limitations
Our study only used respondents from NCR, which may limit its generalisability to other regions of the country where HIV cases are still growing. Furthermore, our study only captured the MSM and TGW populations and did not include other at-risk populations; future studies on other at-risk populations are needed further to validate our theoretical acceptability of HIVST in the Philippines. Furthermore, the conduct of KIIs was done online due to the COVID-19 pandemic, and quarantine protocols and lockdowns were strictly enforced; hence, some reactions to the questions of the participants are not noted due to the unavailability of video. In addition, future investigation is required to identify strategies to support linkage to prevention and treatment services. Finally, quantitative studies may be conducted to triangulate our findings and validate the necessary factors to implement HIVST in the country. These studies would eventually enable the

implementers and key stakeholders to assess the potential public health impact and cost-effectiveness of HIVST in the future.

## CONCLUSION

New HIV testing modalities must be introduced to reach more people in knowing their HIV status and access prevention packages or treatment if needed. This study aimed to understand the acceptability and feasibility of HIVST services as an innovative testing modality to MSMs and TGWs within NCR. The study revealed some important points that need to be prioritised to implement HIVST programmes in the country. First, HIVST must be a part of the national priority to reach more people that the current HIV testing modalities cannot reach. Second, to have a patient-centred HIVST programme in the country and cope with the changing times, the implementers must use and strengthen online platforms because this is an effective way to reach and offer HIVST to the hard-to-reach at-risk populations. Third, the HIVST programme must also be coupled with more mechanisms to ensure patient confidentiality throughout the process. Lastly, standard monitoring, counselling, and linkage to care services must be provided to guarantee the next steps the clients must undertake regardless of their HIVST results. Given all these requirements that need to be considered in the implementation, an HIVST integrated programme in the Philippines will be very promising to engage more people in knowing their HIV status.

**Author affiliations**

[1]LoveYourself Inc, Mandaluyong City, NCR, Philippines
[2]Faculty of Management and Development Studies, University of the Philippines Open University, Laguna, Calabarzon, Philippines
[3]College of Medicine, Pamantasan ng Lungsod ng Maynila, Manila, NCR, Philippines
[4]Institute of Clinical Epidemiology, National Institutes of Health, University of the Philippines Manila, Manila, NCR, Philippines
[5]Department of Clinical Epidemiology, University of the Philippines Manila College of Medicine, Manila, Metro Manila, Philippines

**Acknowledgements** The research team gratefully acknowledges the contributions of many individuals in the study. We would like to thank Mohan Anasalam of Chembio Asia Pacific, Jorelyn Pinuela of Isopharma and Dr. Gundo Weiler of WHO, Philippines for generously giving us the resources to accomplish the study, to John Oliver Corciega, John Darwin Ruanto and Edgar Daniel Bagasol for helping us in the administration and funding of these resources. Many thanks to Paul Victor Junio, Lord-Art Lomarda, Raybert Domingo, Ria Briñes and Kurt Silvano for the creation of promotional materials and online systems to help us gather respondents and organise data. We would also like to thank the LoveYourself Life Coaches and case management team namely Jose Mari Maynes, Emery Mandel, Marlo Bryan Galvez, Tyrone Cudeldiego, Ronald Bugarin, Raymond Martin Manahan, Joy Daguiso, Marvin Frondoza, Roberto Pinauin, Harrold Abilas, Leo Pura, Michael Dela Paz, Aisia Jesse Castelano and Eda Catabas for helping us in addressing the concerns of the participants as they learn more about HIV prevention or treatment packages. The research team would also like to express our gratitude to Mark Ryan Costales, Antonio Latorino, Raphael Kevin Tevar, Queenie Sheena May Mauhay, Reinalyn Espiritu and Lady Lee Callejo for helping us in administering the delivery of necessary materials to the participants. Also, to Rolando Bello Jr. and LoveYourself Rye-ders namely Sir Galo, Reimil, Jayson, Cha, Jenevive, Marco, JR, Dhem, Joel, Fernando, Niel and Caloy for delivering the materials with full confidentiality to our respondents and participants and to Jose Buendia Jr., Richard Bello, Arvin Salayon and AcXess by LoveYourself for taking care of our clients as they visit the facilities to access life-saving HIV treatment. Thank you to our advocates and ambassadors,

Miss Universe 2018 Catriona Gray and Paolo Gumabao for sharing their time and showing their support to HIV advocacy. Lastly, to our 1100 LoveYourself volunteers who helped us in spreading these innovations to the community and to and to all our respondents who generously shared their insights and thoughts to help us create ripples of positive change in the community.

**Contributors** RGP, JDTR and ESB conceptualised the study. MTB, ZJGR, JDTR and ESB developed the protocol. JLGD and JDTR collected the data. JLGD and DPC led the transcription and translation process. JLGD, ZJGR and ESB wrote the first draft of the manuscript. All authors helped with the revision of the manuscript. ESB is responsible for the overall content as the guarantor. All authors agreed on the final version.

**Funding** This study was funded by Global Fund Sustainability of HIV Services for Key Populations in Asia Programme (SKPA Philippines), AIDS Healthcare Foundation–Philippines, Pilipinas Shell Foundation (PSFI), WHO–Representative Office for the Philippines, Joint United Nations Programme on HIV and AIDS–Philippines Country Office, and Chembio Diagnostic Systems.

**Competing interests** None declared.

**Patient and public involvement** Patients and/or the public were not involved in the design, or conduct, or reporting, or dissemination plans of this research.

**Patient consent for publication** Consent obtained directly from patient(s)

**Ethics approval** The Ethics Approval of this study was granted by the University of the Philippines- Manila Research Ethics Board (UPMREB 2019-474-01) before the study started. Online informed consent forms were given to the participants before starting and recording the key informant interviews. The participants were also given a background and the aims of the study, and all procedures were conducted in accordance with the Philippine Data Privacy Act of 2012 (R.A. 10173). Participants gave informed consent to participate in the study before taking part.

**Provenance and peer review** Not commissioned; externally peer reviewed.

**Data availability statement** Data are not available due to the risk of identifying individual participants.

**ORCID iDs**
Jeanno Lorenz G Dinglasan http://orcid.org/0000-0002-5290-8526
John Danvic T Rosadiño http://orcid.org/0000-0002-9885-5300
Denis P Cruz http://orcid.org/0000-0002-9986-8373
Zypher Jude G Regencia http://orcid.org/0000-0002-3383-6336
Emmanuel S Baja http://orcid.org/0000-0002-3888-8880

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
