## [Reviewer comments · BMJ Open]

ARTICLE DETAILS

TITLE (PROVISIONAL)	"Bringing testing closer to you" – Barriers and Facilitators in Implementing HIV Self-Testing among Filipino Men-Having-Sex-with-Men and Transgender Women in National Capital Region (NCR), Philippines: A Qualitative Study
AUTHORS	Dinglasan, Jeanno Lorenz; Rosadiño, John Danvic; Pagtakhan, Ronivin; Cruz, Denis; Briñes, Matthew; Regencia, Zypher Jude; Baja, Emmanuel

VERSION 1 – REVIEW

REVIEWER	Shrivastava, Saurabh R. Shri Sathya Sai Med Coll
REVIEW RETURNED	27-Sep-2021

GENERAL COMMENTS	Remarks: 1. On what criteria, participants were selected as Key Informants?? Did they really justified the clause of being a Key Informant? 2. It was mentioned that the selected participants were grouped into 4 clusters... However, cluster 1 and 3 are the same 3. No mention has been made about obtaining informed consent from them and their privacy 4. The abstract does not provide much information about the barriers to HIVST 5. Limitations of the study needs to be elaborated
---

REVIEWER	Adebayo, Oluwamuyiwa W Pennsylvania State University
REVIEW RETURNED	28-Sep-2021

GENERAL COMMENTS	I appreciate the opportunity to review your manuscript. Below are my comments. 1) Consider spelling out "M" in the estimated statistics for HIV infection 2) There were several grammatical errors. Consider the services of a technical editor to improve the flow and understanding of your manuscript. 3) Is transmission decreased early awareness of testing or early testing and diagnoses? Awareness does not equal engaging in the behavior. 4) What steps were taken to ensure rigor in this study?
--

	5) In Table 1, the classification “Sexual Identity” doesn’t accurately represent TGW. Being transgender is a gender identity and not a sexual identity.
--	---

VERSION 1 – AUTHOR RESPONSE

Responses to the Reviewers

Reviewer: 1

Dr. Saurabh R. Shrivastava, Shri Sathya Sai Med Coll

1. On what criteria, participants were selected as Key Informants? Did they really justify the clause of being a Key Informant?

Response: The selection of the key informants was based on certain criteria. These criteria were included in the manuscript. In addition, the identified key informants were active and supportive members of the LGBTQIA+ community.

2. It was mentioned that the selected participants were grouped into 4 clusters... However, cluster 1 and 3 are the same

Response: This part was revised to make the clustering clearer. The four clusters were (1) MSM who have been tested/screened for HIV at least once, (2) TGW who have been tested/screened for HIV at least once, (3) MSM who were never tested before for HIV, and (4) TGW who were never tested before for HIV for the key informant interviews (KIIs).

3. No mention has been made about obtaining informed consent from them and their privacy.
Response: Obtaining informed consent from the participants was included in the “data collection” section. On the other hand, the privacy and confidentiality statement was included in the “Ethical Considerations and Data Protection” section and the declaration statements after the conclusion.

4. The abstract does not provide much information about the barriers to HIVST

Response: The abstract now included barriers to HIVST.

5. Limitations of the study need to be elaborated.

Response: We provided a separate section in the discussion about the limitations of the study.

Other limitations were also added in the Strength and Limitations section.

Reviewer: 2

Dr. Oluwamuyiwa W Adebayo, Pennsylvania State University

1. Consider spelling out "M" in the estimated statistics for HIV infection.

Response: This was incorporated in the revised manuscript.

2. There were several grammatical errors. Consider the services of a technical editor to improve the flow and understanding of your manuscript.

Response: Thank you. Grammar and spelling were rechecked.

3. Is transmission decreased early awareness of testing or early testing and diagnoses?
Awareness does not equal engaging in the behavior.

Response: We added statements regarding the coupling of activities to engage people in HIV testing and not solely rely on an increase in awareness.

“However, it has to be emphasized that the reduced onward HIV transmission due to awareness in HIV testing must be coupled with rigorous strategies in getting these informed individuals to be engaged in testing. Various strategies may include government activities partnered with community-based organizations, enlisting community leaders for public events, and promotion of HIV testing in facility-based settings and community-based events.”

4. What steps were taken to ensure rigor in this study?

Response: The rigors of the study were ensured by analyzing the data until saturation was reached. This part was included in the data collection section. Another way to ensure rigor was persistent checking of the data analysis by each team member, which was described in the data analysis section.

5. In Table 1, the classification “Sexual Identity” doesn’t accurately represent TGW. Being transgender is a gender identity and not a sexual identity.

Response: The sexual identity was changed to gender identity.